# Examination of the Relationships between Urban Form and Urban Public Services Expenditure in China

**Chunming Bo, Hengzhou Xu and Yong Liu ***

College of Management and Economics, Tianjin University, Tianjin 300072, China; 2015209054@tju.edu.cn (C.B.); xuhengzhou@tju.edu.cn (H.X.)

**\*** Correspondence: yonghliu@tju.edu.cn; Tel.: +86-022-7404-683

**Abstract:** This econometric study contributes to the ongoing debate about the costs and benefits of urban form by employing interdisciplinary means—urban planning, econometrics and public administration—to explore the relationship between urban form and urban public services expenditure. In China, particularly, rapid urbanization is accompanied by an increase of urban public services expenditure and a difference in efficiency, which undermines the promotion of urban public service development. The Chinese government has paid great attention to urban sustainable development and promoting urban public services performance; however, until recently there has been a lack of empirical studies exploring the relationship between urban public services expenditure and urban form. Thus, the present research aims to analyze this issue by using relevant indicators based on an econometric model. The results provide a promising basis for improving urban public services expenditure efficiency. Based on the urban area interpreted by remote sensing data and geographic information system, two urban form metrics, the compactness ratio and the elongation ratio, are selected and quantified to describe urban compactness and urban sprawl accurately. Panel data analyses are performed using a cross-sectional dataset of the 30 cities for the years 2007, 2010 and 2013 to assess the likelihood of association between indicators of urban form and urban public services expenditure, while controlling for other determinants, such as educational level, income per capita, degree of industrialization, and unemployment rate. The results indicate that urban elongation is positively correlated to per capita urban public services expenditure and urban compactness is insignificantly correlated to it. Thus, it is recommended that policymakers consider the relationship between urban form and public services expenditure as part of urban planning and on-going strategies to promote public service efficiency.

**Keywords:** public services expenditure; urban compactness; urban elongation; urban planning; China

## 1. Introduction

Rapid urbanization is accelerating the growth and evolution of economic development and urban form, which is resulting in the increasing of public services expenditure in China. For example, urban public finance spending in China increased to 94,535 billion yuan in 2013. Nevertheless, public services expenditure has not effectively met the needs of urban residents for public resources. The shortage of public services has been a great challenge in China. Moreover, China is characterized by regional differentials, and regional inequalities of public services, particularly high-quality public services. According to a report of the Chinese Academy of Social Sciences (Pan and Wei 2015), only 10% of 288 prefecture-level cities have relatively healthy public services expenditure. Their public services expenditure are unreasonable high and inefficient, which has inhibited the sustainable development of urban public services. Meanwhile, due to the function transition of government from management to services, government at all levels of China is paying increasing attention to establishment of public

services. In that context, it is crucial to explore the relationship between urban form and urban public services expenditure. Therefore, the objective of this research is to discover abovementioned relationship by selecting appropriate indicators and methods.

## 2. Literature Review and Hypothesis Development

Much previous research focuses on the potential impact of urban form on the costs of local public services. The majority of studies of exploring the relationship between urban form and urban public services expenditure can be divided into the following three main categories.

First, urban sprawl increases the costs of public services. Urban sprawl emerged in Europe and the United States in the 1950s (Jackson 1987; Antrop 2000). Although the definition of urban sprawl remains ambiguous, it is generally believed that the characteristics of urban sprawl include low-density, scattered or dispersed strip, and leapfrog development (Burchell et al. 1998; Ewing 2008; Hortas-Rico and Solé-Ollé 2010). For example, the research of Elis-Williams (Elis-Williams 1987) indicates that lower density of individual consumers due to suburbanization undermines economics of scale in the provision of public services, which results in inefficient cost increases. Ewing (Ewing 1997) also discuss the relationship between densities, infrastructure and public services costs, indicating that urban public services costs could be a key source of leverage for urban planners and policymakers to promote more compact urban areas. Employing a cross-sectional dataset of 159 growing counties in the United States from 1982 to 1992, Pendall (Pendall 1999) find that public indebtedness was related to urban sprawl. Similarly, Carruthers (Carruthers 2002) reports that low-density expansion lead to increasing government expenditures on public services because major investment is required to extend highway networks, as well as water, electricity, and sewer lines to a relatively small number of residents. Recently, researchers (Hortas-Rico and Solé-Ollé 2010) employ an empirical analysis of 2500 Spanish municipalities for the year 2003 and a piece-wise linear function to account for the potentially non-linear relationship between sprawl and costs; the findings echo the above mentioned conclusions. Niu, Zhang and Dong note that urban low-density result in a marked increase in public spending using with an analysis of 286 cities for the year 2010 in China (Niu et al. 2013).

In the second strand of literature, compact cities have urban development patterns to promote the cost-effective provision of urban public services. According to Burton (Burton 2000), urban compactness represents relatively high-density, mixed land use and pedestrian-oriented habitation. Cereda notes that the planning strategy of a compact city was helpful to guarantee successful collective transportations, services and provision of public spaces (Cereda 2009). Meanwhile, Coppola et al. design a system of land-use and transport interactions (LUTI) model and applied it to the urban area of Rome to investigate the relationship between sustainability and urban form (Coppola et al. 2014). The results indicate that compact development has better sustainable results than urban sprawl does. In addition, Fernández-Aracil and Ortuño-Padilla (Fernández-Aracil and Ortuño-Padilla 2016) also evaluated the effect of land use patterns on urban public services spending while controlling for other determinants, indicating that compact development help to reduce the costs of provision and maintenance of urban public services.

In the third strand of literature, some studies have indicated that urban sprawl does not always add extra costs to public services. Considering a site-specific focus and the failure to control for the influence of factors might result in controversial conclusions (Frank 1989; Ladd 1998). For example, using cross-sectional data and controlling influencing factors, Ladd and Yinger (Ladd and Yinger 1991) find that public services expenditure increase with density, which is contrary to the finding of previous site-based research. In particularly, according to a "piecewise" regression analysis, Ladd (Ladd 1992) illustrated the relationship might be U-shaped. Richardson and Gordon (Richardson and Gordon 2004) reported that several benefits have been attributed to urban sprawl, such as satisfaction of consumer preferences for single-family housing and improving some problem such as pollution and congestion. However, Ewing and Hamidi (Ewing and Hamidi 2015) believe that every urban development pattern has both costs and benefits.

An assessment of the existing literature informs the theoretical framework of relationship between urban form and urban public services expenditure. Public services expenditure is related to the physical structure of an urban area. Moreover, there is widespread disagreement about the costs and benefits of urban form. Most studies focus on some cities from Europe and the US, and there is relatively little research aimed at investigating the relationship between urban form and urban public services expenditure of developing countries, such as China. Regarding developmental pattern, industrial structure and land use, there are difference between cities of China and developed countries. China is in a period of accelerating urbanization, and at the same time, urban size will expand to that of cities of developed countries, tending to make governments provide more public services. According to the above mentioned conclusions and present situation of China, the authors propose the following hypotheses.

**Hypothesis 1.** *Urban sprawl is positively correlated to per capita urban public services expenditure.*

**Hypothesis 2.** *Urban compactness is negatively correlated to per capita urban public services expenditure.*

The empirical research reviewed in this section can be viewed as an estimation of these hypotheses. Thus, using a cross-sectional dataset of the 30 cities in China for the year 2007, 2010 and 2013, the present research captures the impact of urban form on urban public services expenditure. We believe that a study of the Chinese case will make an effective supplement to the existing literature.

## 3. Methodology

### 3.1. Urban Public Services Expenditure

Public service is characterized by non-excludability and non-rivalry. The former means that if a public service is provided, no one can be excluded from consuming it. The latter means that one person's consumption of public service does not impinge significantly on another's consumption. Public services expenditure in China is mainly focused on strengthening infrastructure construction, creating employment, developing public utility, and publishing public information (Xu 2010). According to the Classification of the Functions of Government published by the International Monetary Fund, financial expenditure mainly contains three elements. The first is expenditure on general public services, which reflects the needs of government and is irrelevant to individual and enterprise consumption. The second is expenditure on economic affairs to manage the economy and improve operational efficiency. The last is people's livelihoods, which means expenditure on services provided directly by the government to society, families and individuals. The detailed categories of purchase of services funded by government are described in Table 1.

**Table 1.** Detailed categories of services.

| Function Classification of Public Expenditure | Specific Categories |
| --- | --- |
| Expenditure on general public services | Transfer payment between governments<br>Defense<br>Public order and safety |
| Expenditure on economic affairs | Agriculture, forestry, fishery, and hunting<br>Fuel and energy<br>Mining, manufacturing, and construction<br>Transportation<br>Communication |
| Expenditure on people's livelihood | Environmental protection<br>Housing and community amenities<br>Health<br>Recreation, culture, and religion<br>Education<br>Social protection |

Source: Ministry of Finance of the People's Republic of China and Classification of the Functions of Government.

Generally, productive government spending cannot finance some services that directly enter into a household's utility functions, which is different from livelihood financial expenditure (Barro 1990). Considering the actual situation of productive government spending in China, there is a limited amount of benefits that resident can obtain from productive government spending are limited (Wang 2014). Therefore, this study excludes expenditure on economic affairs. Meanwhile, we eliminate defense expenditure from the costs of public services because it belongs to central government expenditure (Cheng 2014). According to Wang (Wang 2014), urban public services expenditure is defined as the sum of expenditure subsections, including general public services, public safety, education, and science and technology (see Table 2).

**Table 2.** Description of urban public services expenditure.

| Type | Description |
|---|---|
| General public services | Used to guarantee normal running of organs and institutions, including expenditure for 32 affairs, such as the National People's Congress (NPC), and land resources. |
| Public safety | Sum of preservation of social safety, including expenditure on armed police, national security, law court, etc. |
| Education | Expenditure on educational management, general education, vocational education, etc. |
| Science and technology | Expenditure on fundamental research, application research, technology research and development, etc. |
| Culture, sport, and the media | Expenditure related to culture, cultural relic, sport, radio, film and television, the press and publishing, etc. |
| Social security and employment effort | Expenditure on social security and employment management affairs, civil administration, employment subsidies, social welfare, etc. |
| Medical and health care | Expenditure on medical and health care management affairs, medical service, community health service, etc. |
| Environmental protection | Expenditure on environmental protection management affairs, Environmental monitoring and supervision, pollution prevention, etc. |
| Urban and rural community affairs | Expenditure on urban and rural community affairs management, communal facilities, community housing, etc. |

Source: Ministry of Finance of the People's Republic of China.

### 3.2. Urban Form

Urban form is a spatial pattern related to the pattern of urban development and human activity (Nedovic-Budic et al. 2016). Several indicators represent urban form. For example, Galster et al. (Galster et al. 2001) defined eight distinct dimensions of urban form, namely, density, centrality, clustering, concentration, contiguity, nuclearity, mixed use, and proximity. Similarly, Huang et al. (Huang et al. 2007) employ five urban form indicators. Knaap, Song, and Nedovic-Budic (Knaap et al. 2007) use street network design, land use mix and density as urban form indicators to illustrate the difference of urban development patterns within and across study areas, which partly echo the indicators used by Song (Song 2003). Recognized metrics for measuring urban form present differing interpretations and generally depend on the objects of the study. However, the majority of research that parameterizes urban form tends to focus on compactness and sprawl (Tsai 2005; Wentz 2000; Colaninno et al. 2011), with spatial measures of elongation and compactness being popular choice (Liu et al. 2012). Hortas-Rico (Hortas-Rico and Solé-Ollé 2010) reported that Spanish municipalities with higher levels of urban sprawl lead to greater costs of local public services provision. Fernandez Milan and Creutzig (Fernandez Milan and Creutzig 2016) agreed that sprawled development often translates into larger marginal infrastructure investment than other

development patterns. In addition, concentrated development and plotting the development of urban spatial morphology have always played big roles in urban form.

Which indicators are ultimately used for a given analyses depend on the data available and the objectives of the study. Moreover, existing ideal urban form can almost always be reduced to these two basic development model (Wu and Li 2010).Thus, based on practice and the availability of good-quality and readily accessible data, two indicators directly representing urban compactness and urban sprawl are chosen: the elongation ratio (ER), and the compactness ratio (CR). The ER measures the extended degree of urban area. Webbity put forward the concept of the ER in 1969 with the following formula (Haggett 1997):

$$ER = \frac{L}{L'} \tag{1}$$

where L and L' are the lengths of the long axis and short axis of urban area, respectively. The more extended the urban shape is, the higher the ratio. Meanwhile, Newman and Kenworthy (1989) used urban compactness, which is related density based on molecular and molar measures. Song (Song 2003) develop a series of compactness indicators to estimate the impact of urban growth, including street design and circulation systems, density, land-use mix, accessibility and pedestrian access. Schwarz (Schwarz 2010) parameterizes urban compactness using landscape metrics and population-related indicators. This is similar to Burton (Burton 2002), who described the development of large number of urban compactness indicators. Therefore, the CR is adopted to measure the shape characteristics of a region by adapting the minimum circumscribed circle as a standard, based on the following equation proposed by Cole (Cole 1964):

$$CR = \frac{A}{A'} \tag{2}$$

where A is the urban area, and A' is the smallest circumcircle of the region; the more compact the area is, the higher the ratio.

The urban area is within the urban land boundary using Landsat images and related thematic maps. We employ 30 Landsat TM (thematic mapper) images (2007, 2010) and 30 Landsat OLI (operational land imager) images (2013) to interpret urban land areas, using ENVI 5.3 and ArcGIS 10.2 for data processing. Both automated photo interpretation and visual interpretation are undertaken to identify the urban area of case cities through satellite images and urban administrative boundary maps. During the process, we employ auxiliary information to promote the accuracy of interpretation, such as Google Earth images. Meanwhile, a popular TM band combination of five, four, and three in red, green, and blue color space is used to distinguish the difference between urban land and non-urban land (see Figure 1).

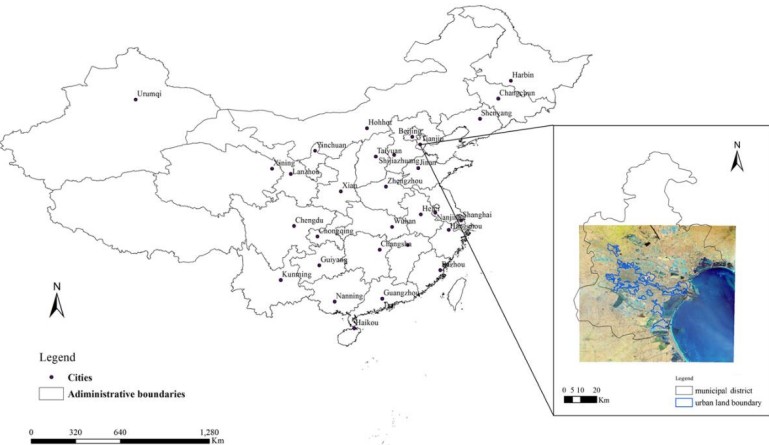

**Figure 1.** Urban boundary interpretation (Tianjin).

*3.3. Panel Data Analysis*

The relationship between urban form and public services expenditure is captured using panel data analyses, which include the cross-sectional and time series dimensions. According to the research of Ladd (Ladd 1992) and Carruthers and Ulfarsson (Carruthers and Ulfarsson 2003), a semi-log form of panel regression models is used. Because of the semi-log form, the estimated coefficients are interpreted as percentage, which means that a unit change in the independent variable corresponds to a percentage change in the dependent variable (Kennedy 1998).

$$\text{Lny}_{it} = \alpha_i + \beta' x_{it} + y' z_{it} + \mu_{it} \tag{3}$$

where $i$ represents the size of the cross section (30 cities), $t$ (2007, 2010 and 2013) represents the dimension of the time series, $\alpha_i$ is a scalar, $\beta'$ and $y'$ are the transpose of $\beta$ and $r$, $\alpha_i$ and $Z_{it}$ are $1 \times K$ vectors of observations of the independent variables (urban form indicators and control variables), and $r_{it}$ is the observation of the dependent variable for individual $i$ at time $t$ (urban public services expenditure). $\mu_{it}$ represents the effects of other factors that are not only unique to individual units but also to time periods, and that can be characterized by an independently and identically distributed random variable with zero mean and variance ($\alpha^2$). In order to more accurately determine the impact of urban form on the costs of providing local public services, it is necessary to control for other potential crucial variables as part of the panel data analysis. Thus, control variables are selected considering the strength of their theoretical and empirical relationships to public services expenditure, as well as available data. The control variables include unemployment rate (Wang 2009), urban land (Bao and Jiang 2016), degree of industrialization, income per capita, gross domestic product (GDP) per capita, urban population (Li 2013), educational level, and government size (Xu 2010). The definition of the control variables on public services expenditure and the expected effect are provided in Table 3.

**Table 3.** Expected influence of the control variables on public services expenditure.

| Variables | Expected Effect | Description |
|---|---|---|
| Unemployment rate | + | The number of unemployed/Labor force |
| Urban land | + | Total number of square meters of land area within entire city according to administrative unit |
| Degree of industrialization | + | Secondary industrial production/Gross domestic product |
| Income per capita | + | Total amount of wages of employees/Total population |
| Educational level | + | Total number of students enrolled at regular institutions of higher education/Total population |
| Government size | + | Total number of employees working in government agencies/Total population |
| Urban population | + | Total population in city-wide area according to administrative unit |
| Gross domestic product (GDP) per capita | + | Total number of GDP/Total population |

The descriptive analysis of selected control variables is presented in Table 4, with data obtained from the China Statistical Yearbooks Database (CSYD) and the National Bureau of China (NBC).

Panel data analysis can have an optimal effect when there are as many sample data (i.e., number of cities in our case) as possible (Hsiao 2014). Steyerberg, Eijkemans, and Habbema (Steyerberg et al. 1999) demonstrate that the selection bias decline as events per predicting variable increase. Peduzzi et al. (Peduzzi et al. 1996) suggest not more than one independent predictor variable per 10 data points, although Vittinghoff and McCulloch (Vittinghoff and Mcculloch 2007) indicate a minimum of 10 outcome events per predictor variables might be too conservative under certain

circumstances. In this study, the sample is 90 (*i* (30 cities) × *t* (three years)), and the maximum number of explanatory variables is 10 (ER, CR, and eight control variables), which is acceptable.

**Table 4.** Descriptive analysis of control variables.

| Variables | | Mean | Maximum | Minimum | Standard Deviation |
|---|---|---|---|---|---|
| *Unemployment (%)* | 2007 | 3.15 | 6.50 | 1.80 | 0.97 |
| | 2010 | 3.19 | 4.20 | 1.30 | 0.77 |
| | 2013 | 2.93 | 4.00 | 1.20 | 0.80 |
| *Urban land (square kilometers)* | 2007 | 15,227.43 | 820 | 2305 | 15,493.70 |
| | 2010 | 15,227.43 | 820 | 2305 | 15,493.70 |
| | 2013 | 15,385 | 824 | 2305 | 15,494.00 |
| *Degree of industrialization (%)* | 2007 | 45.35 | 57.27 | 26.83 | 7.10 |
| | 2010 | 45.05 | 56.17 | 24.00 | 8.00 |
| | 2013 | 44.37 | 55.96 | 22.32 | 8.89 |
| *Income per capita (yuan/person)* | 2007 | 27,891.14 | 49,311.11 | 19,991.83 | 7259.35 |
| | 2010 | 40,786.25 | 71,875.36 | 31,128.48 | 10,606.22 |
| | 2013 | 56,380.37 | 93,996.77 | 43,653.04 | 12,154.55 |
| *GDP per capita (yuan/person)* | 2007 | 34,088.63 | 71,808 | 14,660 | 14,892.95 |
| | 2010 | 50,345.47 | 103,625 | 25,622 | 18,625.16 |
| | 2013 | 84,162.53 | 185,338 | 37,691 | 37,939.88 |
| *Educational level (%)* | 2007 | 5.70 | 11.02 | 1.28 | 2.47 |
| | 2010 | 6.54 | 12.55 | 1.71 | 2.74 |
| | 2013 | 7.32 | 12.61 | 2.11 | 3.05 |
| *Government size (%)* | 2007 | 1.22 | 2.22 | 0.58 | 0.39 |
| | 2010 | 1.23 | 2.37 | 0.63 | 0.37 |
| | 2013 | 1.36 | 3.10 | 0.74 | 0.59 |
| *Total population (1000 people)* | 2007 | 723.57 | 3235.32 | 148.79 | 568.32 |
| | 2010 | 750.24 | 3303.45 | 158.80 | 582.06 |
| | 2013 | 770.21 | 3358.40 | 163.20 | 588.29 |

## 4. Results and Discussion

### 4.1. Results

Government expenditures on public services in 2007, 2010, and 2013 for the selected 30 Chinese cities are presented in Table 5. The growth trend of per capita urban public services expenditure is obvious. The average values of per capita urban public services expenditure in 2007, 2010 and 2013 are 3224.90, 5272.22 and 8611.32, respectively. The urban public services expenditure per capita of most cities ranges from 1000 yuan to 5000 yuan in 2007 (26 cities) and 2010 (21 cities), with only two (2007) and six (2010) cities ranging from 5001 yuan to 10,000 yuan, and two and three cities having values above 10,001. Moreover, in 2013, the urban public services expenditure per capita of 21 cities ranges from 5001 yuan to 10,000 yuan, with three cities ranging from 1000 yuan to 5000 yuan, and six cities having values above 10,001 yuan.

The calculation results of urban CR and ER for 30 cities in 2007, 2010, and 2013 are presented in Table 6. Urban CR is increasing and urban ER is gradually decreasing (the average value of urban compactness in 2007, 2010 and 2013 is 0.184, 0.197 and 0.207, respectively, and the average value of urban elongation in 2007, 2010 and 2013 is 2.10, 1.98 and 1.95, respectively). There are more cities with a CR ranging from 0.16 to 0.30 in 2007 (15 cities) and 2013 (18 cities) than in 2010 (14 cities). There are also more cities with an ER ranging from 1.51 to 2.50 in 2007 (18 cities) compared with 2010 (17 cities) and 2013 (13 cities). Moreover, the number of cities with a CR ranging from 0.00 to 0.15 is on the decline (12, 10, and 9 in 2007, 2010 and 2013, respectively). The number of cities with an ER ranging from 1.00 to 1.50 has risen (8, 8, and 12 in 2007, 2010 and 2013, respectively). From this description and Table 5, we can see that Chinese cities are undergoing compact development and the urban form of every city changes dynamically. However, there is still relatively large upside potential to realize concentrated urban area considering that there is no city in China with a calculated CR above 0.40.

**Table 5.** Urban public services expenditure per capita.

| Category | Cities |
|---|---|
| Per capita total expenditure (2007) (yuan) | |
| 1000–5000 | Shijiazhuang; Taiyuan; Hohhot; Shenyang; Changchun; Harbin; Nanjing; Hangzhou; Hefei; Fuzhou; Nanchang; Jinan; Zhengzhou; Wuhan; Changsha; Nanning; Haikou; Chongqing; Chengdu; Guiyang; Kunming; Xian; Lanzhou; Xining; Yinchuan; Urumqi |
| 5001–10,000 | Tianjin; Guangzhou |
| 10,001–25,000 | Beijing; Shanghai |
| Per capita total expenditure (2010) (yuan) | |
| 1000–5000 | Shijiazhuang; Taiyuan; Changchun; Harbin; Hefei; Fuzhou; Nanchang; Jinan; Zhengzhou; Wuhan; Changsha; Nanning; Haikou; Chongqing; Chengdu; Guiyang; Kunming; Xian; Lanzhou; Xining; Urumqi |
| 5001–10,000 | Hohhot; Shenyang; Nanjing; Hangzhou; Guangzhou; Yinchuan |
| 10,001–25,000 | Beijing; Tianjin; Shanghai |
| Per capita total expenditure (2013) (yuan) | |
| 1000–5000 | Shijiazhuang; Nanning; Harbin |
| 5001–10,000 | Taiyuan; Hohhot; Shenyang; Changchun; Hangzhou; Hefei; Fuzhou; Nanchang; Jinan; Zhengzhou; Wuhan; Changsha; Haikou; Chongqing; Chengdu; Guiyang; Kunming; Xian; Lanzhou; Xining; Yinchuan |
| 10,001–25,000 | Beijing; Tianjin; Shanghai; Nanjing; Guangzhou; Urumqi |

**Table 6.** Urban form descriptors for 30 Chinese cities during the years 2007, 2010 and 2013.

| Category | Cities |
|---|---|
| Compactness ratio (2007) | |
| 0.00–0.15 | Beijing; Tianjin; Shijiazhuang; Changchun; Harbin; Wuhan; Guangzhou; Chongqing; Guiyang; Xian; Lanzhou; Urumqi |
| 0.16–0.30 | Shenyang; Shanghai; Hangzhou; Hefei; Fuzhou; Nanchang; Jinan; Nanjing; Hohhot; Zhengzhou; Nanning; Haikou; Chengdu; Xining; Yinchuan |
| 0.31–0.40 | Taiyuan; Changsha; Kunming |
| Compactness ratio (2010) | |
| 0.00–0.15 | Tianjin; Shijiazhuang; Changchun; Harbin; Wuhan; Guangzhou; Chongqing; Xian; Lanzhou; Urumqi |
| 0.16–0.30 | Beijing; Taiyuan; Hohhot; Shenyang; Shanghai; Nanjing; Hangzhou; Fuzhou; Nanchang; Jinan; Zhengzhou; Nanning; Haikou; Chengdu; Guiyang; Kunming; Xining; Yinchuan |
| 0.31–0.40 | Hefei; Changsha |
| Compactness ratio (2013) | |
| 0.00–0.15 | Beijing; Tianjin; Shijiazhuang; Changchun; Harbin; Nanjing; Wuhan; Lanzhou; Urumqi |
| 0.16–0.30 | Hohhot; Shenyang; Fuzhou; Nanchang; Jinan; Zhengzhou; Changsha; Guangzhou; Nanning; Chongqing; Chengdu; Guiyang; Xian; Xining |
| 0.31–0.40 | Taiyuan; Shanghai; Hangzhou; Hefei; Haikou; Kunming; Yinchuan |

**Table 6.** *Cont.*

| Category | Cities |
|---|---|
| | Elongation ratio (2007) |
| 1.00–1.50 | Beijing; Shenyang; shanghai; Hangzhou; Wuhan; Changsha; Chengdu; Kunming |
| 1.51–2.50 | Tianjin; Shijiazhuang; Taiyuan; Hohhot; Changchun; Harbin; Nanjing; Hefei; Fuzhou; Nanchang; Guiyang; Zhengzhou; Guangzhou; Nanning; Haikou; Xian; Yinchuan; Urumqi |
| 2.51–11.00 | Jinan; Chongqing; Lanzhou; Xining |
| | Elongation ratio (2010) |
| 1.00–1.50 | Beijing; Shanghai; Hangzhou; Hefei; Wuhan; Changsha; Chengdu; Kunming |
| 1.51–2.50 | Tianjin; Taiyuan; Hohhot; Shenyang; Changchun; Harbin; Nanjing; Fuzhou; Nanchang; Guangzhou; Nanning; Haikou; Chongqing; Guiyang; Xian; Yinchuan; Urumqi |
| 2.51–11.00 | Shijiazhuang; Jinan; Zhengzhou; Lanzhou; Xining |
| | Elongation ratio (2013) |
| 1.00–1.50 | Beijing; Changchun; Shanghai; Hangzhou; Hefei; Wuhan; Changsha; Guangzhou; Kunming; Yinchuan; Urumqi; Chengdu |
| 1.51–2.50 | Tianjin; Taiyuan; Hohhot; Shenyang; Harbin; Fuzhou; Nanchang; Zhengzhou; Nanning; Haikou; Guiyang; Chongqing; Xian |
| 2.51–11.00 | Shijiazhuang; Nanjing; Jinan; Lanzhou; Xining |

To select the appropriate model used in the panel data analysis, F-tests, redundant fixed effects (RFE) test, the Hausman test and the Breusch-Pagan and Lagrangian multiplier (BP-LM) test are employed. The result of model tests using the Eviews 9 and Stata 14 software are presented in Table 7.

**Table 7.** Model test results: redundant fixed effects (RFE) test; Breusch-Pagan and Lagrangian multiplier (BP-LM) test.

| | | Statis | d.f. | Chi-sq | d.f. | Prob. |
|---|---|---|---|---|---|---|
| RFE | F | 3.85 | (23,29) | - | - | 0.00 |
| | Chi-square | 88.20 | 23 | - | - | 0.00 |
| Hausman | Random | - | - | 50.45 | 10 | 0.00 |
| BP-LM | P > chibar2 = 0.1558 | - | - | - | - | - |

Note: Statis represents statistic. d.f. represents degree of freedom. Chi-sq represents Chi-square statistic. prob. represents probility.

The RFE test indicate that the pooled model is better than the fixed effects model ($p$-value > 0.05) (Hausman 1978). The Hausman test indicates that the random effects model is better than the fixed effects model ($p$-value > 0.05) (Hausman 1978). Lastly, the BP-LM test indicate that the pooled model is better than the random effects model (Prob > chibar2 = 0.1445 > 0.05) (Breusch and Pagan 1980). Therefore, a fixed effects model would be best for investigating the relationship between urban form and urban public services expenditure (Table 8).

**Table 8.** Results of panel data analysis.

| Total Panel (Unbalanced) Observations: 63 | | | | |
|---|---|---|---|---|
| **Variable** | **Coefficient** | **Std. Error** | **t-Statistic** | **Prob.** |
| *Urban compactness* | 0.30 | 0.78 | 0.38 | 0.70 |
| *Urban elongation* | 0.25 ** | 0.09 | 2.72 | 0.01 |
| *Educational level* | 0.17 ** | 0.04 | 4.32 | 0.00 |
| *GDP per capita* | $1.86 \times 10^{-6}$ | $1.95 \times 10^{-6}$ | 0.95 | 0.35 |
| *Income per capita* | $2.18 \times 10^{-5}$ ** | $4.87 \times 10^{-6}$ | 4.48 | 0.00 |
| *Degree of industrialization* | 0.03 ** | 0.01 | 2.44 | 0.02 |
| *Urban land* | $3.28 \times 10^{-5}$ | $5.82 \times 10^{-5}$ | 0.56 | 0.58 |
| *Government size* | 0.44 ** | 0.18 | 2.44 | 0.02 |
| *Unemployment rate* | 0.12 | 0.06 | 1.98 | 0.06 |
| *Urban population* | 0.00 | 0.00 | −0.40 | 0.69 |
| C | 2.83 ** | 1.16 | 2.44 | 0.02 |
| **Effects Specification** | | | | |
| **Cross-section fixed (dummy variables)** | | | | |
| R-squared | 0.98 | Mean dependent variance | | 8.47 |
| Adjusted R-squared | 0.95 | S.D. dependent variance | | 0.68 |
| S.E. of regression | 0.14 | Akaike information criterion | | −0.73 |
| Sum squared residual | 0.60 | Schwarz criterion | | 0.43 |
| Log likelihood | 57.04 | Hannan-Quinn criterion. | | −0.28 |
| F-statistic | 40.33 | Durbin-Watson statistic | | 2.63 |
| Prob (F-statistic) | 0 | | | |

Note: C is constant coefficient.

Table 8 present the results of panel data analysis for the relationship between urban form and urban public services expenditure. Urban ER has a significant positive correlation with urban public services expenditure with a coefficient of 0.25, but urban compactness is insignificantly correlated to it. Given that a semi-log is adopted by taking the log of the dependent variables only, the estimated parameters are interpreted as percentages. Then, a unit increase in urban ER increases urban public services expenditure per capita by 0.25%. Moreover, educational level, income per capita, degree of industrialization, government size, and constant manifest positive correlation with urban public services expenditure. GDP per capita, urban land, unemployment rate and urban population are unrelated to urban public expenditure.

## 4.2. Discussions

The results provide support for Hypothesis 1 that the urban ER has a significant positive correlation with urban public services expenditure. Urban sprawl is associated with low population density, which might result in a series of unnecessary government spending. In other words, urban sprawl might undermine economies of scale in the provision of public services. For example, the medical and educational systems have significant infrastructure built in fixed places, which might be amortized over more people at higher densities (Ewing 1997). In addition, poor accessibility to urban services due to urban sprawl possibly has induced the inefficient performance of public services. However, overcrowding due to high density might produce additional costs of public services such as traffic congestion and, thus, the crowding effect of high-density areas needs to be considered by policymakers and planners when trying to shape a reasonable urban form. Moreover, urban sprawl is associated with difficulty in promoting universities and research and development (R&D) companies to concentrate together and form agglomeration economies. On the other hand, compact of urban space will probably strengthen cooperation and communication between companies and universities, which could save transaction costs, improve economic efficiency and promote scientific and technological levels. Agglomeration economies can realize the effective configuration of resource

automatically and, thus, can be accompanied by the savings of science and technology expenditure. The research of Keller (Keller 2002) supports these results mentioned above. Meanwhile, Xu, Wang, and Tan (Xu et al. 2007) consider that declining geographical distance between enterprises has contributed to technological knowledge sharing and the growth of industrial clusters. Moreover, the geography of concentration of cultural industries can produce positive temporal and spatial spill over among cities (Kwanwai and Mok 2014).

Additionally, urban sprawl might reduce social interaction and contribute to socioeconomic segregation (Downs 1999; Brueckner 2000). Then, the adverse consequences of increasing crime rate and inequity of fiscal resources will arise in low-income neighborhoods, accompanied by high public services expenditure. Moreover, excessive land conversion to urban use diminishes the extent of farmland and forests (Club 1998), which might lead to declining ability to purify air and extra costs in providing certain local services. Similarly, according to Ewing et al. (Ewing et al. 2008), high-density development helps people live within walking or bicycling distance of some of their daily destinations. A pedestrian-oriented development pattern, not an automobile-dependent one, reduces greenhouse gas emissions, which would help save costs related to environment protection. Although the relationship between urban compactness and urban public services expenditure is insignificant, urban sprawl is unfavorable to save costs in the provision of public services. The results indicate that urban form in China has undergone rapid dynamic changing from 2007 to 2013, which mainly stemmed from the two pathways of urbanization: passive and active urbanization. Passive urbanization is the phenomenon by which government stimulates population from rural to urban areas by way of farmland expropriation (Yu et al. 2013), which would increase urban sprawl or elongation as well as increase the number of unemployed. Active urbanization occurs when rural residents/farmers spontaneously move to a city owing to the urban industrialization and economic development, increasing either compaction or elongation, depending on whether the settlement is built mostly in a central or outlying area.

Regarding control variables, government size, educational level, income per capita, and the degree of industrialization are significantly correlated to urban public services expenditure. Government size is the important contributing factor of urban public services expenditure. An increase in the size of the government means that massive public services expenditure is devoted to government operations and salaries of government officials. The positive coefficient of educational level and urban public services expenditures indicates increasing public services attached to overall improvement of education. Income is also an influencing factor in the demand for public services. According to Wagner's Law, increasing income enables residents to focus on the quality of life, which would be expected to increase urban public services expenditure (Narayan et al. 2012). Similarly, the improvement of industrialization promotes economic development and, in turn, increases residents' demand for public services.

## 5. Conclusions and Suggestions

The present study undertakes a quantitative analysis of the relationship between specific indicators of urban form and urban public services expenditure. While controlling for such variables as educational level, income per capita, degree of industrialization, and urban land, and so forth, the results indicate that urban elongation could be a contributing factor in urban public services expenditure in China. Although the role of urban public services expenditure is not obviously influenced by the urban CR, the compact city is a sustainable urban form for China. Successful urban development should prioritize the provision of high-quality public services and minimize public services expenditure. Moreover, the impact of passive urbanization could increase urban elongation, and, in turn, increase urban public services expenditure. Thus, changes in urban planning to minimize the negative effects of increased urban elongation are necessary. For example, promoting active urbanization could be a strategy to reduce urban public services expenditure in China.

However, urban compaction might result in higher density, leading to traffic congestion and air pollution (Troy 1996; Rudlin and Falk 1999). The current Chinese urban development pattern is typical

of high urban population density, automobile orientation and high rate of industrial aggregation. Therefore, there is a complex trade-off between advantages of increasing compactness and increasing population density. According to our conclusion, urban spatial extension undermines cost-effective public service provision. However, there are too many negative effects of compactness such as traffic congestion in some Chinese urban areas, and various economic and environmental problems are caused by increased compactness. Therefore, policy making should identify an optimum degree of urban compactness to maintain sufficient public service provision to offset the negative effects.

Some limitations of this research are worth mentioning. The indicators for measuring urban form and urban public services expenditure are limited. Therefore, the research provides only an empirical correlation of appointed aspects. Nonetheless, the explorative research provides a starting point for further investigation of the impact of urban form on urban services expenditure in China.

**Acknowledgments:** The authors thank the reviewers and the editor for their helpful comments, which have significantly improved the quality of the paper.

**Author Contributions:** Chunming Bo wrote the draft paper and Yong Liu and Xu reviewed the paper. Yong Liu also structured the paper.

**Conflicts of Interest:** The authors declare no conflict of interest.

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
