# Peer review of "Examination of the Relationships between Urban Form and Urban Public Services Expenditure in China"

_admsci, doi:10.3390/admsci7040039_

Round 1

Reviewer 1 Report

This manuscript examines the relationship between urban form and urban public services expenditure in China.  The authors find that urban elongation is positively correlated with urban public services expenditure per capita, and recommend that urban planners consider the relationship considered in the study. 

The article is well-written, and presents a compelling research question.  However, the manuscript would require significant revisions before it can be brought to publication in this journal. 

General Comments

- I would encourage the authors to use the present tense (i.e. not the past tense) throughout the manuscript.  At the very least, it should be consistent. 

- Overall, the manuscript is well-written.  The authors demonstrate very strong competency in English language and style, but a good editing would still be needed before publication.  In my remarks below, I only make reference to the spelling and grammatical changes that are the most significant. 

Abstract

- The abstract could be strengthened by briefly explaining the two urban form metrics used in the study.

- In general, the abstract should contain stronger language with regard to the significance of the study, with references to other studies in the literature.  This is sometimes the only thing that the reader will read, so it should be clear what is new. 

1. Introduction

- Page 1, Line 27: I would recommend changing “cannot” to “has not.”

- Page 1, Line 32: Change “inhabited” to “inhibited.” 

- Page 1, Line 37: Delete the word “would.” 

2. Literature Review

- The authors note in the first paragraph that the literature can be divided into three categories, but only make explicit reference to one category (the first).  The remaining two categories should be clearly identified in the text. 

- I agree that China would make a compelling supplement to the existing literature, but are there really no studies that have applied this research question, or least an aspect of this research question, to China?

- The authors reference several studies that focus on Europe and the US, but China likely presents a very different institutional context as compared to both cases.  To what extent do those cases apply (or not) to China?  What are the authors’ hypotheses?  Regardless of whether China is similar or different from Europe and/or the US, the paper requires a much deeper theoretical foundation as to the relationship between urban form and urban public services expenditure in this case.

3.  Methodologies

3.1 Urban public services expenditure 

- The first paragraph requires deeper explanation of some of the key concepts discussed; for example, non-excludable, non-rivalry, and livelihood financial expenditure.

- Further explanation of the definition of urban public services expenditure is needed.  Is this definition supported by the literature?  In particular, I have strong concerns that national defense is included as an urban public expenditure, when it’s typically regarded as a national policy. 

3.2 Urban form

- The authors reference several notable studies in this area.  However, they don’t clearly explain the choice of the two metrics.  The authors correctly note that there are many dimensions and indicators of urban form, but it’s critically important in this literature to explain why these particular metrics were chosen. 

- Following my previous thought, I’m curious as to why the authors selected two metrics that are quite old, when the literature has produced better metrics more recently, especially in the studies that they reference. 

3.2 Panel data analysis

- Sub-header label should be changed to 3.3

- Following my last comment on the literature review, why does this econometric model hold a theoretical relationship with China?  This is why a clear statement of hypotheses is needed, so as to make a clear arc between economic theory and the model itself. 

- Variables such as degree of industrialization, educational level, and government size need explanation/definition.  Both their numerators and denominators are unclear.

4. Results and discussion

4.1 Results

- Clear interpretation of the variables (on pg. 7) and coefficients (on pg. 8) is needed. 

4.2 Discussions

- In this section, the authors finally provide a theoretical basis for the study, but these types of hypotheses and relationships should be presented much earlier in the manuscript, as noted previously.

- In general, the causal language in this section is far too strong, especially since the authors note at the end of the manuscript that these results are limited and preliminary. 

5. Conclusion and suggestions

- Page 9, Line 263: Change “per capital” to “per capita.”

Author Response

Dear Editor:

I would like to re-submit the attached manuscript entitled “Examination of the relationships between urban form and urban public services expenditure in China.” The manuscript ID is admsci-218670.

The manuscript has been carefully rechecked and the appropriate changes have been made in accordance with the referee’s suggestions. The responses to the referee’s comments have been prepared and attached herewith.

Reviewers' Comments:

1. I would encourage the authors to use the present tense (i.e. not the past tense) throughout the manuscript. At the very least, it should be consistent

Authors’ reply:

Thank you for your comments. We have used the present tense throughout the manuscript.

2. Abstract

- The abstract could be strengthened by briefly explaining the two urban form metrics used in the study.

- In general, the abstract should contain stronger language with regard to the significance of the study, with references to other studies in the literature. This is sometimes the only thing that the reader will read, so it should be clear what is new.

Authors’ reply:

Thank you for your advice. We have added to the description of the two urban form metrics and the significance of the study in a      bstract.

3. Introduction

- Page 1, Line 27: I would recommend changing “cannot” to “has not.”

- Page 1, Line 32: Change “inhabited” to “inhibited.”

- Page 1, Line 37: Delete the word “would.

Authors’ reply:

Thank you for these recommendations. We have revised them accordingly.

4. Literature Review

- The authors note in the first paragraph that the literature can be divided into three categories, but only make explicit reference to one category (the first).  The remaining two categories should be clearly identified in the text.

Authors’ reply:

Thank you for pointing this out. We have identified and explained the remaining two categories. Please refer to the Literature Review.

- I agree that China would make a compelling supplement to the existing literature, but are there really no studies that have applied this research question, or least an aspect of this research question, to China?

Authors’ reply:

Thank you for your questions. Yes, you are correct; some studies have addressed this research question. We have added related literature in China of the relationship between urban form and urban public services expenditure, for example Niu, zhang, and Dong (2013).

- The authors reference several studies that focus on Europe and the US, but China likely presents a very different institutional context as compared to both cases. To what extent do those cases apply (or not) to China? What are the authors’ hypotheses? Regardless of whether China is similar or different from Europe and/or the US, the paper requires a much deeper theoretical foundation as to the relationship between urban form and urban public services expenditure in this case.

Authors’ reply:

Thank you for your advice, Which we added paragraphs to explain. Please refer to the last two paragraphs of Section 2, the Literature Review and Hypothesis Development..

5. Methodologies

Urban public services expenditure

- The first paragraph requires deeper explanation of some of the key concepts discussed; for example, non-excludable, non-rivalry, and livelihood financial expenditure.

Authors’ reply:

Thank you for this practical advice. We have added the explanation of non-excludable, non-rivalry, and livelihood financial expenditure and we have divided government expenditure into three categories, namely general public services expenditure, economic affairs expenditure and people’s livelihood expenditure (refer to the beginning of Subsection 3.1).

- Further explanation of the definition of urban public services expenditure is needed. Is this definition supported by the literature? In particular, I have strong concerns that national defense is included as an urban public expenditure, when it’s typically regarded as a national policy.

Authors’ reply:

Thank you for mention this. Yes, the research of Wang supports the definition of urban public services expenditure. We have expanded the definition of urban public services expenditure and national defense is excluded from urban public expenditure according to related literature (e.g., Cheng 2014; Wang 2014).

3.2 Urban form

- The authors reference several notable studies in this area. However, they don’t clearly explain the choice of the two metrics.  The authors correctly note that there are many dimensions and indicators of urban form, but it’s critically important in this literature to explain why these particular metrics were chosen.

- Following my previous thought, I’m curious as to why the authors selected two metrics that are quite old, when the literature has produced better metrics more recently, especially in the studies that they reference.

Authors’ reply:

Thank you for your comments. Considering the review’s comment, we have explained the choice of the two metrics and cited related literature in the revised version of the paper (refer to the second paragraph of Subsection 3.2).

3.2 Panel data analysis

- Sub-header label should be changed to 3.3

Authors’ reply:

Thank you for your advice. We have revised the label.

- Following my last comment on the literature review, why does this econometric model hold a theoretical relationship with China? This is why a clear statement of hypotheses is needed, so as to make a clear arc between economic theory and the model itself.

Authors’ reply:

Thank you for your questions. We have added the hypotheses about the relationship between urban form and urban public services expenditure. (Please refer to section 2)

- Variables such as degree of industrialization, educational level, and government size need explanation/definition. Both their numerators and denominators are unclear.

Authors’ reply:

Thank you for pointing this out. We have added a definition of the variables (refer to Table 3).

4. Results and discussion

4.1 Results

- Clear interpretation of the variables (on pg. 7) and coefficients (on pg. 8) is needed.

Authors’ reply:

Thank you for your advice. We have added an interpretation of the variables and coefficients (refer to Subsection 4.1).

4.2 Discussions

- In this section, the authors finally provide a theoretical basis for the study, but these types of hypotheses and relationships should be presented much earlier in the manuscript, as noted previously.

Authors’ reply:

Thank you for pointing this out. We have added the hypotheses much earlier in the revised version of the paper, as explained in previous response.

- In general, the causal language in this section is far too strong, especially since the authors note at the end of the manuscript that these results are limited and preliminary.

Authors’ reply:   

Thank you for mentioning this . We have diluted the causal language to reflect the fact that the results are limited and preliminary.

5. Conclusion and suggestions

- Page 9, Line 263: Change “per capital” to “per capita.”

Authors’ reply:

Thank you for pointing this out. We have revised this to “per capita”.

I thank the referee for the thoughtful suggestions and insights, which have enriched the manuscript and produced a more balanced and better account of the research. I hope that the revised manuscript is now suitable for publication in Administrative Sciences.

I look forward to your reply.

Reviewer 2 Report

General

This paper addresses an important issue in urban planning, local government and public economics, namely the potential relationship between urban form and public service costs/spending, and brings forward new evidence from a very important case study country, China – important because of the extent of recent urbanisation there. It contains some useful data and analysis, but it suffers from several weaknesses in the way in which this is contextualised, discussed and interpreted. It would benefit from rewriting significantly in parts to try to address these weaknesses.

The main weaknesses are

More contextualisation needed in terms of international comparison and also in terms of the process of urbanisation in China

Failure to discuss important conceptual issues, including dynamics of urban public service spending in context of growth (e.g. distinction between capital and current spending), economics of scale in urban size and public services, a proper account of the different types of urban public services beyond classic ‘public good’ definition

Casual discussion of ‘efficiency’ without acknowledging that, in the absence of measures of service output, standards or quality, you can say nothing about efficiency from just looking at expenditure per capita. Failure to fully develop the important point that larger /denser cities can afford to develop more specialised and sophisticated services, and it becomes worthwhile to do so, so they spend more.

Methodology – no explanation of why such a small sample of cities (30) were selected for study, nor how they were selected, nor any data on how representative they are.

Style of tabular presentation in some cases is not easy to read or follow.

The discussion of the ‘panel’ regression analysis is technocratic and misleading – this is basically just a pooled cross –sectional analysis.

There is a confused and inconsistent approach to what functions of local government are included for consideration. At one point it is stated that the ‘economic development’ function is excluded from consideration, yet there is later nearly a whole paragraph discussing an aspect of this (agglomeration and higher education/research). Furthermore, it is surprising that urban transportation is not discussed as one of the key urban services, since this is the most important one which the massive international literature on urban sprawl is most concerned about. It is unclear whether this is seen as ‘economic’ or ‘social/environmental’ – surely it is both. Similarly all the other network infrastructure services – energy, water, sewerage/drainage, telecoms – are they included or excluded? These services are arguably the ones which are most affected by urban sprawl.

Specific

Line 27  Just quoting one figure (95bn yuan) here is meaningless without benchmark comparators, trends over time for China or from other countries.

33  You can’t say they are unreasonably high or inefficient on the data presented, which include no evidence on service outputs or standards

59-60  How come Niu et al can analyse 286 cities but you are only doing 30?

69  I don’t think you have defined sustainability. Obviously sustainability is a different concept from cost/expenditure – how would you relate them?

76  Are Ladd and Yinger holding service standards constant? I doubt it.

83-5 Inadequate theoretical/conceptual discussion here e.g. differen types of urban service.  Fiscal federalism?

90  Need to discuss dynamics, leads/lags/retrofit, capital vs current etc. see above general comment.

94-98  I think it is a bit simplistic to reduce it to only two dimensions. A lot of the literature on urban form and sustainability distinguishes more than two dimensions e.g. mixed use, network connectivity  see also 147-175.

105 should read ‘Public goods are characterized….’ [public goods defined in this way are a narrower subset of public services, many of which are not non-excludable nor non-rival]

124 See general comment 7. There are clearly issues about urban form and efficiency in relation to services which may be labelled ‘economic affairs’.

129 Table 2. Are all of the services here equally delegated to local government , and to an equal degree in different cities?  In most countries science and technology would be seen as a higher level national function, as would parts of some other services. Why is transport not in this table (see general comment 7)

153 This measure seems remarkably crude. I would have thought measures based on network (e.g. road) length relative to population etc. would be more sophisticated version….

180  I don’t think three data points really constitutes a time series in the normal full sense of the term. For this reason I think this is not a panel in the full sense of the terms.

200 Table 3  Two issues (a) you have two scale variables competing in this model, urban population and urban land. That is questionable.
(b) Government size variable is verging on the spurious – it is surely just a corollary of spending more on services

This is the point where lack of theoretical or literature based discussion of economics of urban size and scale seems very obvious. What is the role of scale variables. Why are they not entered in non-linear form, e.,g. quadratic.?  Similarly dynamics of scale – there is no growth rate control variable.

213  No information about why 30 cities nor why/how  these 30 were selected and how representative they are.

235 this statistics of 0.40 is meaningless without international benchmarks

Tables 5 & 6 I don’t like this way of presenting this information. I would prefer a straightforward table showing the cities with their values for the 3 years, ranked in a meaningful size order.

238-250 see general comment 6. I don’t think dressing this up as a panel analysis really washes.

264  You can’t say anything about efficiency!

265-277  see comment 7 – inconsistent to include this discussion of an economic development issue

290-293 – interesting general observations here, but I would expect citation of references to support.

319  you have not addressed traffic congestion, nor presented any data on it, nor (apparently) analysed expenditure on roads/transit, so far as I can see.

Author Response

[25 Sep 2017]

Dear Reviewer:

I would like to re-submit the attached manuscript entitled “Examination of the relationships between urban form and urban public services expenditure in China.” The manuscript ID is admsci-218670.

The manuscript has been carefully rechecked and the appropriate changes have been made in accordance with the referee’s suggestions. The responses to the referee’s comments have been prepared and attached herewith.

Reviewers' Comments:

1. Abstract

- Page 1, Line 6: I recommend beginning the abstract with: “This econometric study…”

Authors’ reply:

Thank you for your comments. We have revised them accordingly.

2. - Page 1, Line 7: The use of the term “interdisciplinary means” is unclear.

Authors’ reply:

Thank you for your advice. We have added urban planning, econometrics and public administration as the description of interdisciplinary means.  

3. - I still recommend including stronger language with regard to the significance of the study, with references to other studies in the literature.  My suggestion is to take the last three sentences of the introduction – starting with “The Chinese government has paid great attention to…” – and move them to the abstract.  That text captures the tone that I’m referring to.

Authors’ reply:

Thank you for pointing this out. According to your advice, we have moved above mentioned sentences to the abstract. And we added the function transition of government to emphasize the significance of the study.

4. Introduction

- Page 1, Line 32: Change “hare” to “have”

 Literature Review and Hypothesis Development

- Page 2, Line 44: Change first sentence to present tense.

- Page 2, Line 62: Change to “second” (not “Second”).

- Page 2, Line 83: I recommend replacing “indicates” with “informs”

Authors’ reply:

Thank you for these recommendations. We have revised them accordingly.

5. Panel data analysis

- Table 3: Check the definition of the unemployment rate – It’s typically defined as Unemployed/Labor Force.

Authors’ reply:

Thank you for your questions. You are correct. We have revised it.

6. Results

- I recommend adding two lines that explain the coefficients for urban compactness and urban elongation presented in Table 8. The authors explain how to interpret a semi-log form in Section 3.3.  It would be helpful for the reader to make that connection in this section, since they’re the key results of the study. 

Authors’ reply:

Thank you for this practical advice. We have added the interpretation of the coefficient for urban elongation. And considering that urban compactness is insignificantly correlated to urban public services expenditure, we don’t explain the coefficient for urban compactness.

I thank the referee for the thoughtful suggestions and insights, which have enriched the manuscript and produced a more balanced and better account of the research. I hope that the revised manuscript is now suitable for publication in Administrative Sciences.

I look forward to your reply.

Round 2

Reviewer 1 Report

Overall, these revisions represent a significant improvement to the manuscript.  I therefore recommend the following edits before publication. 

Abstract

- Page 1, Line 6: I recommend beginning the abstract with: “This econometric study…”

- Page 1, Line 7: The use of the term “interdisciplinary means” is unclear.

- I still recommend including stronger language with regard to the significance of the study, with references to other studies in the literature.  My suggestion is to take the last three sentences of the introduction – starting with “The Chinese government has paid great attention to…” – and move them to the abstract.  That text captures the tone that I’m referring to. 

1. Introduction

- Page 1, Line 32: Change “hare” to “have”

2. Literature Review and Hypothesis Development

- Page 2, Line 44: Change first sentence to present tense.

- Page 2, Line 62: Change to “second” (not “Second”).

- Page 2, Line 83: I recommend replacing “indicates” with “informs”

3.3 Panel data analysis

- Table 3: Check the definition of the unemployment rate – It’s typically defined as Unemployed/Labor Force.

4.1 Results

- I recommend adding two lines that explain the coefficients for urban compactness and urban elongation presented in Table 8.  The authors explain how to interpret a semi-log form in Section 3.3.  It would be helpful for the reader to make that connection in this section, since they’re the key results of the study.  

Author Response

[25 Sep 2017]

Dear Reviewer:

I would like to re-submit the attached manuscript entitled “Examination of the relationships between urban form and urban public services expenditure in China.” The manuscript ID is admsci-218670. The paper was coauthored by

The manuscript has been carefully rechecked and the appropriate changes have been made in accordance with the referee’s suggestions. The responses to the referee’s comments have been prepared and attached herewith.

Reviewers' Comments:

1. Abstract

- Page 1, Line 6: I recommend beginning the abstract with: “This econometric study…”

Authors’ reply:

Thank you for your comments. We have revised them accordingly.

2. - Page 1, Line 7: The use of the term “interdisciplinary means” is unclear.

Authors’ reply:

Thank you for your advice. We have added urban planning, econometrics and public administration as the description of interdisciplinary means.  

3. - I still recommend including stronger language with regard to the significance of the study, with references to other studies in the literature.  My suggestion is to take the last three sentences of the introduction – starting with “The Chinese government has paid great attention to…” – and move them to the abstract.  That text captures the tone that I’m referring to.

Authors’ reply:

Thank you for pointing this out. According to your advice, we have moved above mentioned sentences to the abstract. And we added the function transition of government to emphasize the significance of the study.

4. Introduction

- Page 1, Line 32: Change “hare” to “have”

 Literature Review and Hypothesis Development

- Page 2, Line 44: Change first sentence to present tense.

- Page 2, Line 62: Change to “second” (not “Second”).

- Page 2, Line 83: I recommend replacing “indicates” with “informs”

Authors’ reply:

Thank you for these recommendations. We have revised them accordingly.

5. Panel data analysis

- Table 3: Check the definition of the unemployment rate – It’s typically defined as Unemployed/Labor Force.

Authors’ reply:

Thank you for your questions. You are correct. We have revised it.

6. Results

- I recommend adding two lines that explain the coefficients for urban compactness and urban elongation presented in Table 8. The authors explain how to interpret a semi-log form in Section 3.3.  It would be helpful for the reader to make that connection in this section, since they’re the key results of the study. 

Authors’ reply:

Thank you for this practical advice. We have added the interpretation of the coefficient for urban elongation. And considering that urban compactness is insignificantly correlated to urban public services expenditure, we don’t explain the coefficient for urban compactness.

I thank the referee for the thoughtful suggestions and insights, which have enriched the manuscript and produced a more balanced and better account of the research. I hope that the revised manuscript is now suitable for publication in Administrative Sciences.

I look forward to your reply.